# Resistin Induces Migration and Invasion in PC3 Prostate Cancer Cells: Role of Extracellular Vesicles

**DOI:** 10.3390/life13122321

**Published:** 2023-12-10

**Authors:** Mario Israel Oregel-Cortez, Héctor Frayde-Gómez, Georgina Quintana-González, Victor García-González, Jose Gustavo Vazquez-Jimenez, Octavio Galindo-Hernández

**Affiliations:** 1Departamento de Bioquimíca, Facultad de Medicina, Universidad Autónoma de Baja California, Mexicali 21100, Baja California, Mexico; mario.israel.oregel.cortez@uabc.edu.mx (M.I.O.-C.); fraydeh@uabc.edu.mx (H.F.-G.); quintanag@uabc.edu.mx (G.Q.-G.); vgarcia62@uabc.edu.mx (V.G.-G.); 2Laboratorio Multidisciplinario de Estudios Metabólicos y Cáncer, Universidad Autónoma de Baja California, Mexicali 21100, Baja California, Mexico; 3Facultad de Deportes, Universidad Autónoma de Baja California, Mexicali 21289, Baja California, Mexico; 4Hospital Regional de Especialidad No. 30, Instituto Mexicano del Seguro Social, Mexicali 21100, Baja California, Mexico; 5Laboratorio de Fisiología, Departamento de Fisiología, Facultad de Medicina, Universidad Autónoma de Baja California, Mexicali 21100, Baja California, Mexico; gustavo.vazquez@uabc.edu.mx

**Keywords:** resistin, prostate cancer, extracellular vesicles, FAK, invasion

## Abstract

Resistin is an adipokine with metabolic and inflammatory functions. Epidemiological and translational studies report that an increase in plasma levels and tissue expression of resistin increases the aggressiveness of prostate tumor cells. Extracellular vesicles (EVs) are secreted constitutively and induced by cytokines, growth factors, and calcium and are found in multiple biological fluids such as saliva, serum, semen, and urine. In particular, EVs have been shown to promote tumor progression through the induction of proliferation, growth, angiogenesis, resistance to chemotherapy, and metastasis. However, the role of resistin in the migration, invasion, and secretion of EVs in invasive prostate tumor cells remains to be studied. In the present study, we demonstrate that resistin induces increased migration and invasion in PC3 cells. In addition, these phenomena are accompanied by increased p-FAK levels and increased secretion of MMP-2 and MMP-9 in resistin-treated PC3 cells. Interestingly, EVs isolated from supernatants of PC3 cells treated with resistin induce an increase in migration and invasion accompanied by high MMP-2 and MMP-9 secretion in an autocrine stimulation model. In summary, our data for the first time demonstrate that resistin induces migration and invasion, partly through the secretion of EVs with pro-invasive characteristics in PC3 cells.

## 1. Introduction

Prostate cancer is a global public health problem. According to the Global Cancer Observatory, in 2020, more than 1 million new diagnoses were reported, resulting in the deaths of more than 375,000 men worldwide [1,2,3]. Prostate cancer is caused by mutations in tumor suppressor genes and oncogenes such as p53, Src, FAK, and Akt, resulting in uncontrolled growth and proliferation [4]. Although prostate cancer can originate from any prostate cell, 95% of tumors originate from luminal epithelial cells [5].

Evidence links obesity with the development of multiple types of cancer [6] because adipose tissue can secrete different molecules that can regulate metabolism [7]. In addition, obesity is associated with various pathologies such as hypertension, diabetes, and prostate cancer [8]. Evidence suggests that at least 20% of all tumors are caused by excess weight, causing a more aggressive disease [9]. Adipose tissue releases more than 20 different types of adipokines, including leptin, adiponectin, TNF-α, and resistin, which can promote tumor growth, proliferation, and metastasis [10]. In obesity, the paracrine communication between adipocytes and cells of the immune system is altered due, among other factors, to the deregulation of adipokine secretion, leading to low-grade chronic inflammation. The increase or decrease in the secretion of hormones, growth factors, and adipokines derived from adipose tissue promotes tumor growth, proliferation, migration, and invasion, with subsequent metastasis. In particular, in prostate cancer, the adipokines leptin, visfatin, and resistin have been related as promoters of various stages of tumor progression. However, the molecular mechanisms involved have not been fully elucidated [11,12].

Resistin (an adipose tissue-specific secretory factor) is a protein rich in cysteine residues and was originally proposed as a link between obesity and diabetes. In humans, this adipokine is found in blood circulation (2–40 ng/mL) as an oligomer and is widely expressed and secreted by macrophages infiltrating visceral adipose tissue. In contrast, adipocytes and preadipocytes express minimal protein levels [13]. Structurally, resistin has several subunits that are held together by non-covalent interactions. This characteristic allows the origination of a multimeric complex consisting of hexamer-forming disulfide bonds. Resistin-mediated transduction pathways have been described in various cancer cells, including signaling through toll-like receptor 4 (TLR4) and activation of the PI3K/Akt/NFκB pathway. Activation of these pathways can cause a chronic inflammatory state to be maintained through the constant secretion of pro-inflammatory cytokines. This can lead to a favorable tumor microenvironment. Cytokines, in turn, can activate specific pathways such as MAPKs and JAK/STAT that will promote cellular responses such as tumor proliferation [13,14]. Likewise, resistin induces the activation of PI3K concomitant with the phosphorylation and activation of Akt, resulting in increased cell proliferation in prostate cancer cells [15].

Migration and invasion are associated with cancer pathophysiology, giving rise to metastasis. Migration is associated with cell motility promoted by the participation of proteins such as Arp2/3, WASP, and Cdc42 [10]. Integrins are transmembrane receptors that couple the extracellular matrix with the cytoskeleton. Therefore, they promote cell survival and migration and promote the activation/phosphorylation of FAK at the Y397 residue, which is associated with the metastasis process [16]. Additionally, FAK overexpression and amplification are associated with tumor progression, favoring migration, invasion, metastasis, and a poor prognosis. Gelatinases are enzymes that degrade extracellular matrix components, such as collagen. It has been observed that the overexpression and increase in the secretion of MMP-2 and MMP-9 are associated with processes such as invasion, metastasis, and a poor prognosis, resulting in key factors in tumor progression [17,18].

Extracellular vesicles (EVs) are a group of vesicles bounded by a lipid bilayer and secreted by multiple cell types, including exosomes, microvesicles, and apoptotic bodies. EVs secreted by tumor cells have been related to different stages of tumor progression, such as proliferation, migration, invasion, angiogenesis, metastasis, and resistance to infection and chemotherapy [19]. EV secretion is a constitutive process present in multiple cell types. However, exogenous stimuli such as growth factors, epinephrine, and adenosine diphosphate induce an increase in secretion and the differential expression of associated molecules in EVs from different cancer cell lines. The cargo molecules present in EVs, including proteins, lipids, miRNA, mRNA, and DNA, are responsible for promoting the different stages of tumor development [20]. Therefore, the composition and function of EVs as intercellular information packets depend on various factors present in the tumor microenvironment (cytokines, hormones, and growth factors, among others) [19,20].

In this work, we show that treatment with resistin induces an increase in FAK phosphorylation levels and an increase in MMP-2 and MMP-9 secretion in PC3 cancer cells. In addition, resistin promotes increased migration and invasion of PC3 cells. Interestingly, EVs isolated from supernatants from PC3 stimulated with resistin promote migration and invasion in a model of autocrine cell communication. Therefore, our data for the first time indicate that resistin induces increased migration and invasion in PC3 cells. This is accompanied by the secretion of EVs with cargo molecules that induce processes associated with metastasis.

## 2. Materials and Methods

### 2.1. Chemicals

Resistin, FAK antibody (sc-271126), anti-phospo-FAK antibody (Tyr-397, sc-81493), anti-CD63 antibody (sc-5275), anti-actin antibody, and agarose were obtained from Santa Cruz Biotechnology (Santa Cruz, CA, USA). Anti-Flotillin-2 antibody and basement membrane matrix (BD Matrigel) were obtained from BD Biosciences (Bedford, MA, USA).

### 2.2. Cell Culture

The PC3 prostate cancer cells were cultured in 100 mm Petri dishes with Dulbecco’s Modified Eagle’s Medium (DMEM) supplemented with 3.7 g/L sodium bicarbonate, 10% fetal bovine serum (FBS), and antibiotics in a humidified atmosphere containing 5% CO_2_ and 95% air at 37 °C. For experimental purposes, PC3 cells were starved in DMEM without FBS for 18 h before treatment with resistin.

### 2.3. Stimulation of PC3 Cancer Cells with Resistin

After starvation, 8 × 10^6^ PC3 cells (confluent cultures) were washed twice with phosphate-buffered saline (PBS), re-fed in DMEM without FBS, and then untreated or treated with resistin at various concentrations and times indicated. The stimulation was terminated by aspirating the conditioned medium.

### 2.4. Isolation of EVs from Conditioned Medium of PC3 Cells Stimulated with Resistin

PC3 cells were maintained and cultured in DMEM with 10% FBS (with serum EVs). Once the cultures reached 80–90% confluence (8 × 10^6^ cells/dish), they were starved by removing this DMEM-FBS and replacing it with DMEM only (without FBS) for 18 h. At the end of this period, the cells were treated with and without resistin using only DMEM (without FBS) medium for 48 h. Therefore, our experimental conditions do not include the presence of FBS. Conditioned medium (supernatant) was collected from PC3 cells that were unstimulated and stimulated with resistin (25 ng/mL). Isolation of EVs was performed as described previously [21,22,23]. Briefly, the conditioned medium was centrifuged twice for 15 min at 200× *g*. Supernatants were then sequentially centrifuged at 600× *g* twice for 30 min, once at 2000× *g* for 30 min, once at 10,000× *g* for 30 min, and once at 120,000× *g* for 75 min. Finally, the EVs were washed once with PBS 1X at 120,000× *g* for 75 min. After centrifugation at 120,000× *g*, the supernatant was discarded, and the pellet was resuspended in 1X PBS (≈100 µL/condition). The EV fractions were enriched in exosomes and microvesicles. Finally, the protein levels in the EV fraction were analyzed using the micro-Bradford protein assay.

### 2.5. Treatment of PC3 Cells with EV Fractions Isolated from Supernatants of PC3 Cells Stimulated with Resistin

Confluent cultures of PC3 cells were washed twice with PBS, re-fed in DMEM for 30 min, and then stimulated with EV fractions obtained from 8 × 10^6^ PC3 cells unstimulated or stimulated with resistin (25 ng/mL). Each condition was treated with 30 µg/mL of EVs. EVs stimulation was terminated by aspirating the medium, and/or cells were solubilized in 0.5 mL of ice-cold RIPA buffer (1.5 mM MgCl_2_, 1 mM EGTA, 150 mM NaCl, 10 mM sodium pyrophosphates, 1 mM sodium orthovanadate, 100 mM NaF, 10% glycerol, 1% Triton X-100, 1% sodium deoxycholate, 0.1% SDS, 1 mM PMSF, 50 mM HEPES, pH 7.4). The micro-Bradford protein assay analyzed the protein concentration of each sample.

### 2.6. Western Blotting

Proteins (≈20 µg/condition) were processed by PAGE under denaturing conditions with 10% gels and transferred to nitrocellulose membranes or, alternatively, PVDF. Membranes were stained with Ponceau red to confirm protein transfer to the membrane. Next, the membranes were blocked with 5% non-fat dried milk in PBS pH 7.2/0.1% Tween 20 (wash buffer) and incubated with the indicated primary antibody at 4 °C overnight. The membranes were washed three or four times with the washing buffer for 15 min and incubated with the secondary antibody (horseradish peroxidase-conjugated) at room temperature. After incubation, the membranes were washed to remove excess antibodies and visualize the bands with ECL detection reagent. Autoradiograms were scanned or photographed with the ChemiDoc XRS+. The bands were analyzed with the ImageJ software (1.54c version, NIH, Bethesda, MD, USA), while the statistical analysis was performed with GraphPad Prism (version 8.0.2, 263, January 2019).

### 2.7. Scratch Wound Assay

Cultures of PC3 cells (100% confluent) were exposed to 12 μM mitomycin C to block cell proliferation.

Cell cultures were scratch-wounded using a sterile 200 μL pipette tip, washed twice with PBS, and re-fed with DMEM stimulated or unstimulated with resistin or EV fractions from 8 × 10^6^ PC3 cells stimulated or unstimulated with resistin. The progress of cell migration into the wound was photographed at the end of the experiment (48 h) using an inverted microscope coupled to a camera. Each experimental condition was repeated three times.

### 2.8. Zymography

Confluent cultures of PC3 cells were treated with resistin or EV fractions obtained from 8 × 10^6^ PC3 cells unstimulated or stimulated with 25 ng/mL resistin for 48 h (experimental condition), and conditioned medium was collected and concentrated using centricon filters (Millipore, Burlington, MA, USA). An equal volume of non-heated conditioned medium samples was mixed with 5X non-reducing sample buffer (2.5% SDS, 1% sucrose, and 4 μg/mL phenol red) without reducing agent and loaded onto 8% acrylamide gels copolymerized with gelatin at 1 mg/mL. Gels were rinsed thrice in 2.5% Triton X-100 and then incubated in incubation buffer (50 mM Tris-HCl pH 7.4 and 5 mM CaCl_2_) at 37 °C for 48 h. Gels were fixed and stained with a staining solution (0.25% Coomassie Brilliant Blue G-250 in 10% acetic acid and 30% methanol). The gels were washed twice with the destaining solution (40% methanol and 10% acetic acid) to remove the excess stain. Proteolytic activity was detected as clear bands against the background stain of the undigested substrate.

### 2.9. Invasion Assays

Invasion assays were performed by the modified Boyden chamber method in 24-well plates containing 12 cell culture inserts with 8 μm pore size (Costar, Corning Inc., Corning, NY, USA). Briefly, 30 μL of BD Matrigel was added to culture inserts and incubated for 30 min at 37 °C. Cells were plated at 1 × 10^5^ per insert in 100 μL serum-free DMEM in the top chamber. The lower chamber contained 600 μL DMEM with resistin (25 ng/mL) or EV fractions from PC3 cells unstimulated or stimulated with resistin. After 48 h of incubation, PC3 cells and the Matrigel were removed from the upper surface with cotton swabs, while the cells that invaded were fixed with 100% methanol for 5 min. Next, the cells were stained with 0.1% crystal violet in PBS. The dye was eluted with 500 μL of 10% acetic acid, and the absorbance at 600 nm was measured. Background values were obtained from wells without cells.

### 2.10. Spheroid Growth Assay (3D Culture)

The development of the spheroid assay was based on the protocol of Saraiva et al. [24], with some modifications. Briefly, 24-well plates were coated with 1.5% agarose dissolved in 1X PBS (400 μL/well). PC3 cell cultures (90% confluence) were trypsinized, and the cell suspension (1 × 10^5^ cells/well) was placed on the agarose-coated wells. The spheroids were cultured under the indicated conditions (DMEM with 5% FBS alone or DMEM supplemented with 25 ng/mL resistin + 5% FBS) for seven days, changing the fresh medium every 48 h. During the experiment, photos were taken on days 3, 5, and 7 with a camera-coupled microscope. The spheroidal area was analyzed with the ImageJ software (Bethesda, MD, USA; 1.54c version). For each time point, the areas of four spheroids were averaged. Statistical analyzes were performed using Student’s *t*-tests.

### 2.11. Statistical Analysis

Results are expressed as mean ± S.D. Data were statistically analyzed using one-way ANOVA and Newman–Keuls’s multiple comparison test. A statistical probability of *p* < 0.05 was considered significant.

## 3. Results

### 3.1. Resistin Induces an Increase in the Migration in PC3 Cancer Cells

Resistin overexpression has been reported in high-grade tumor tissue compared with low-grade lesions and benign prostatic hyperplasia [15,25]. High-grade lesions are associated with a high metastatic capacity, involving cell migration and invasion [15,25]. Therefore, we decided to evaluate the effect of resistin on cell migration in PC3 cells by scratch wound assay. PC3 cell cultures were scratch-wounded and stimulated at different resistin concentrations (10, 25, 50, and 100 ng/mL) and 10% FBS as a positive control for 48 h. As observed in Figure 1A, resistin promotes an increase in the migration of PC3 cells, with a maximum migration at 25 ng/mL of resistin. Since a high migratory capacity of PC3 cells treated with resistin at 25 ng/mL was observed, we decided to use this condition for all subsequent experiments. FAK is a non-receptor tyrosine kinase that is overexpressed in prostate cancer and regulates processes such as migration and invasion by modulating the dynamics of focal adhesions. Therefore, we decided to evaluate the effect of resistin on the levels of Tyr-397 phosphorylation of FAK, indicative of maximum catalytic activity [26]. Cell lysates from PC3 cells stimulated with and without resistin at 25 ng/mL at the indicated times were obtained and processed by Western blotting with anti-p-FAK (Tyr-397), anti-FAK, and anti-actin antibodies as loading controls. Our data indicate that treatment of PC3 cells with resistin induces a time-dependent increase in FAK phosphorylation levels, with a maximum phosphorylation level at 60 min of treatment (Figure 1B). As an inhibitor of cell proliferation, they were pre-treated with mitomycin C (Appendix A).

### 3.2. Resistin Promotes an Increase in the Invasion in PC3 Cells

MMP-2 and MMP-9 are gelatinases involved in the invasion and metastasis of cancer cells, and their synthesis and secretion are regulated by pathways such as MAPK and FAK/Src. In line with this notion, we determined the impact of resistin on the secretion of MMP-2 and MMP-9. Briefly, confluent PC3 cell cultures were treated with or without 25 ng/mL resistin for 48 h, at which time the conditioned medium was obtained and processed by zymography. As shown in Figure 2A, resistin-treated PC3 cells significantly increase the secretion of MMP-2 and MMP-9 in conditioned medium. High expression and secretion of MMP-2 and MMP-9 have been associated with increased prostatic tumor invasion, an important phenomenon for subsequent tumor metastasis [27,28]. Based on previous data indicating that resistin induces the secretion of MMPs, the role of resistin in tumor invasion was evaluated. The invasion assay was performed by the Boyden chamber method. The inserts were coated with Matrigel, and 1 × 10^5^ cells/well were placed, while in the lower chamber was added DMEM supplemented or not with resistin and FBS as a positive control. Our data indicate that resistin induces a significant increase in the invasive capacity of PC3 cells (Figure 2B).

### 3.3. Resistin Induces an Increase in 3D Culture Spheroid Growth in PC3 Cells

Our previous data show that resistin increases PC3 cell invasion. Therefore, we decided to examine the role of resistin in spheroid formation (3D cultures) in PC3 cells. PC3 cells were seeded in 24-well plates coated with 1.5% agarose. PC3 cells (1 × 10^5^ cells/well) were treated with 5% FBS or resistin (25 ng/mL) + 5% FBS for 7 days. Finally, photos were taken on days 3, 5, and 7. As shown in Figure 3, our data indicate that resistin + 5% FBS induces a significant growth of the spheroids, doubling the area of the spheroid on day 7 of treatment with respect to the condition with 5% FBS alone.

### 3.4. Resistin Induces the Secretion of EVs That Promote PC3 Cell Migration

EVs are vesicles that contain multiple cargo molecules, which are determined according to the cell type secreting the EVs and the particular signals to which the cell is exposed [29,30]. Multiple factors, such as physical signals, hypoxia, low glucose levels, cytokines, and growth factors, induce EV secretion. Therefore, we decided to analyze whether resistin modulates the function of EVs in PC3 cells. First, we obtained a fraction enriched in EVs by differential centrifugation, according to the strategy schematized in Figure 4A. To verify that the EV fractions were properly obtained, they were processed by Western blotting using anti-CD63 and anti-flotillin-2 antibodies since these proteins are considered EV markers. The results show that we obtained an EVs fraction positive for flotillin-2 and CD63, indicating that these fractions are enriched in EVs (Figure 4B). EVs have been involved in tumor progression, participating in the modulation of the tumor microenvironment and favoring tumor growth, proliferation, migration, and invasion. However, the role of resistin in EV secretion remains to be studied. Thus, we decided to evaluate the effect of resistin on the secretion of EVs. PC3 cultures were treated without or with resistin for 48 h, a conditioned medium was obtained, and EV fractions were isolated (Figure 4A). We decided to study whether the EV fractions from PC3 cells treated with resistin regulate cell migration. Our data indicate that EVs derived from resistin-treated PC3 cells induce an increase in PC3 cell migration compared to migration promoted by EVs from PC3 cells not stimulated with resistin (Figure 4C). In this assay, we included control of uptake inhibition with EVs pre-treated with annexin V [31,32,33], a protein that binds to phosphatidylserine on EVs and regulates uptake. This indicates that EVs stimulate cell migration.

### 3.5. EVs Derived from Resistin Treatment Induce Cell Invasion

MMP-2 and MMP-9 are involved in the remodeling process of the extracellular matrix, increasing its secretion during invasion and tumor metastasis. Therefore, we determined whether EVs from PC3 cells regulate the secretion of MMPs. PC3 cultures were treated for 48 h with EVs isolated from the supernatant of PC3 cells treated with or without resistin. Subsequently, the supernatants were collected and analyzed by zymography. Consistent with our previous data, resistin induces the secretion of MMP-2 and MMP-9 compared to the control group (Figure 5A). In addition, treatment of PC3 cells with DMEM EVs induces the secretion of MMP-2 and MMP-9 to a lesser extent. Interestingly, treating PC3 cells with resistin EVs promotes the secretion of MMP-2 and MMP-9 without reaching the levels of DMEM EVs. We included EVs pre-incubated with annexin V, which significantly decreased the secretion of MMP-9 into the conditioned medium without affecting MMP-2 levels (Figure 5A). We decided to evaluate the EVs by zymography. Our data suggest that EVs carry MMP-9, with a slight increase in the EVs resistin fraction; while MMP-2 is not detectable in both fractions. As controls we added the conditioned medium and the free fraction of EVs. These data suggest that MMP-9 is an cargo enzyme in PC3 cell-derived EVs (Appendix A). Since the secretion of MMPs in the extracellular space is strongly associated with cancer invasion, we decided to evaluate the impact of EVs isolated from PC3 cells on cell invasion by Boyden chamber assays. The inserts were coated with Matrigel, placing PC3 cells on this extracellular matrix surface. In the lower chamber, EVs isolated from PC3 cells treated or not with resistin were placed. As shown in Figure 5B, EVs isolated from resistin-treated PC3 cells induce a significant increase in cell invasion compared to PC3 cells stimulated with EVs isolated from PC3 cells without resistin treatment or DMEM alone (control group).

## 4. Discussion

Resistin is an adipokine involved in metabolic and inflammatory processes. Studies have indicated a relationship between high levels of circulating resistin with overweight and obesity. These conditions are associated with the development of diabetes, hypertension, dyslipidemia, metabolic syndrome, and an increased risk of cancer [34,35]. There is evidence that resistin increases aggressiveness in breast cancer, favoring processes such as tumor growth by increasing BCL-2 and BCL-xL activity, accompanied by a decrease in cleaved caspase-7 and -3. Interestingly, resistin promotes breast cancer metastasis by increasing Src activity and phosphorylation and favoring PKCα translocation to the nucleus [36]. Resistin has been linked to tumor progression in patients with prostate cancer. In a study in which Korean patients participated, it was observed that as the Gleason score increased, there was an increase in the expression of resistin in prostate tumor tissue. This was accompanied by an increase in the proliferation and phosphorylation of Akt in the prostate cancer cell lines PC3 and DU-145 [15]. Therefore, we decided to evaluate whether resistin impacts other cellular processes associated with prostatic cancer progression. Here, we show that resistin induces an increase in migration and invasion in PC3 cells, concomitant with an increase in the secretion of MMP-2 and MMP-9 and high levels of FAK phosphorylation (Tyr-397) [15,35]. FAK overexpression (and high phosphorylation levels) is associated with increased tumor aggressiveness in processes such as migration, invasion, and metastasis [37,38,39]. It has been shown that the phosphorylation of FAK at its Tyr-397 residue is related to its maximum catalytic activity, activating proteins such as PI3K, Src, Grb7, and N-WASP [40,41]. Resulting in a high migratory and invasive capacity in cancer cells [41].

The secretion of MMP-2 and MMP-9 by cancer cells is an important mechanism of the cell invasion process since these enzymes degrade basal membrane components such as type IV collagen, a process associated with highly invasive and metastatic tumors [24]. The secretion of MMPs can be induced by the activation of the PI3K/Akt pathway, which leads to the activation of the transcription factor NF-κB. The transcription factor NFκB regulates the expression of multiple proteins that modulate cell invasion, among which are Snail1 and 2, Twist1 and 2, vimentin, MMP-9, and MT1-MMP, which induces the activation of MMP-2 [24,25]. Our group confirmed that exposure of PC3 cells to resistin increases the secretion level of MMP-2 and MMP-9 in the conditioned medium, favoring invasion in prostate cancer cells.

EVs are involved in cellular communication processes as well as physiological processes. EVs contain in their membrane phospholipids, transmembrane proteins, and proteins that are components of lipid rafts. At the same time, inside the EVs, there are various intracellular proteins, second messengers, and genetic material that can be packaged in the EVs [42,43]. As a consequence, the properties and biological role of EVs may differ depending on the parent cell and the microenvironment to which they are exposed. EVs can fuse with the plasma membrane of target cells, transferring genetic information, second messengers, and receptors that can induce cell signaling, whereby the recipient cell acquires new functions [42,43]. The secretion of EVs by cancer cells represents a mechanism by which bioactive molecules such as nucleic acids, chemokine receptors, growth factor receptors, functional transcription factors, enzymes, and various intracellular proteins that modulate signaling pathways are transferred. In addition, EVs are secreted in response to a wide variety of stimuli, released at high concentrations by cancer cells. Thus, we hypothesized that resistin could regulate cell–cell communication mediated by EVs in the PC3 cells. Our results indicate that resistin induces the secretion of EVs that promote migration and invasion in PC3 cells, simulating a model of autocrine communication. Resistin-mediated EVs secretion can be supported by the fact that, in MDA-MB-231 invasive mammary cancer cells, resistin promotes an increase in intracellular calcium concentration, a phenomenon known to induce EV secretion [42,43]. EVs contain a variety of cargo molecules that promote invasion and metastasis, including PI3K/AKT, αvβ5 integrin, EGFRvIII, and a wide variety of miRNAs [36,37]. This strongly suggests that resistin-exposed PC3 cells secrete EVs with cargo molecules that induce invasion and metastasis, highlighting the importance of resistin in prostatic tumor progression. However, further assays are required to support these events. Our data indicate that the EVs-enriched fraction presents Flot-2 and CD63; however, it is necessary to characterize the EVs populations by electron microscopy and/or nanoparticle tracking analysis.

Resistin is a protein secreted by cells such as macrophages, and it regulates the inflammatory response by promoting the secretion of TNF-alpha and IL-12 in macrophages [44]. These effects could be induced by their possible interaction with and activation of TLR4 [42,45]. It has been observed that the activation of TLR4 generates an increase in calcium release [46,47] from the endoplasmic reticulum to the cytoplasm (intracellular calcium). In cancer cells, resistin promotes phosphorylation of Src (a downstream target of FAK kinase) and increases intracellular calcium levels [42]. It has been shown that the FAK/Src pathway induces Akt activation in cancer cells. This results in increased secretion of MMP-2 and MMP-9, favoring proliferation, migration, invasion, and metastasis [48]. Particularly, an increase in intracellular calcium levels is a key event in microvesicle and exosome biogenesis and secretion by tumor cells [43]. In line with this notion, we hypothesize that, in PC3 cells, resistin interacts with and activates TLR4, increasing p-FAK and intracellular calcium levels. FAK activation promotes the phosphorylation of p-Src and the PI3K/Akt/NF-kB pathway, which increases the secretion of MMP-2 and MMP-9, favoring proliferation, migration, and invasion in PC3 cells. Furthermore, the increase in intracellular calcium levels generates an increase in the EV biogenesis and secretion of cargo molecules that induce the invasion process of PC3 cells.

In summary, our data show for the first time that resistin triggers invasion in PC3 cells. In addition, PC3 cells exposed to resistin secrete EVs that induce migration and invasion in PC3 cells. This highlights resistin as an important adipokine in prostate cancer progression and metastasis.

## Figures and Tables

**Figure 1 life-13-02321-f001:**
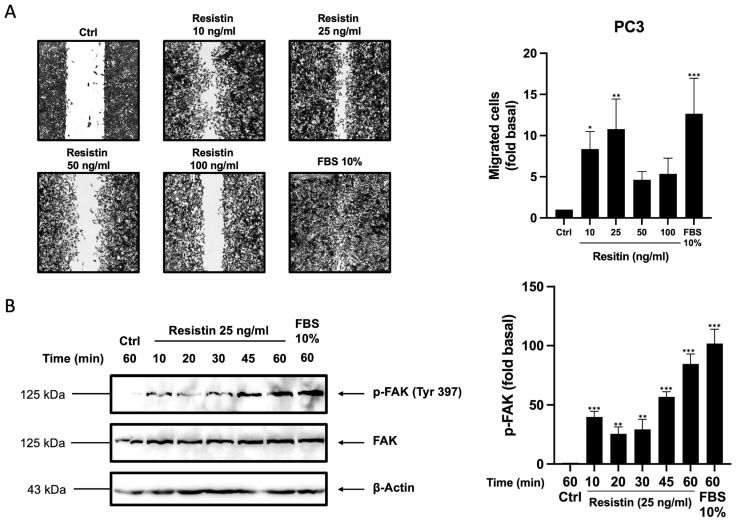
Resistin induces migration in prostate cancer cells. (**A**) Cell migration was evaluated through a scratch wound assay. Cells were treated with different concentrations of resistin, and the migration of cells was analyzed after 48 h. (**B**) Cell lysates of PC3 cells treated with or without resistin were analyzed with Western blotting to evaluate p-FAK and FAK levels and actin as a loading control. The graphs represent the mean ± SD of three independent experiments and are expressed as the fold of migration above the control value. Asterisks indicate comparisons made to control. * *p* < 0.05, ** *p* < 0.01, and *** *p* < 0.001.

**Figure 2 life-13-02321-f002:**
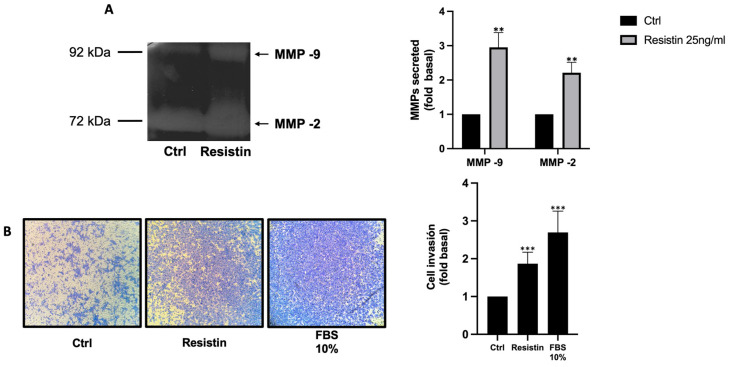
Resistin increases PC3 cell invasiveness by upregulating MMP-2 and MMP-9 secretion. (**A**) PC3 cells were treated for 48 h with resistin, at which time a conditioned medium was obtained and processed through zymography to evaluate MMP-2 and MMP-9 secretion; (**B**) Boyden chamber method was used to evaluate PC3 cell invasion. Cells were seeded in the upper chamber precoated with Matrigel, and treatment with resistin was added to the lower chamber. After 48 h, cell invasion was analyzed. The graph represents the mean ± SD of three independent experiments and is expressed as the fold of invasion above the control value. ** *p* < 0.01, and *** *p* < 0.001.

**Figure 3 life-13-02321-f003:**
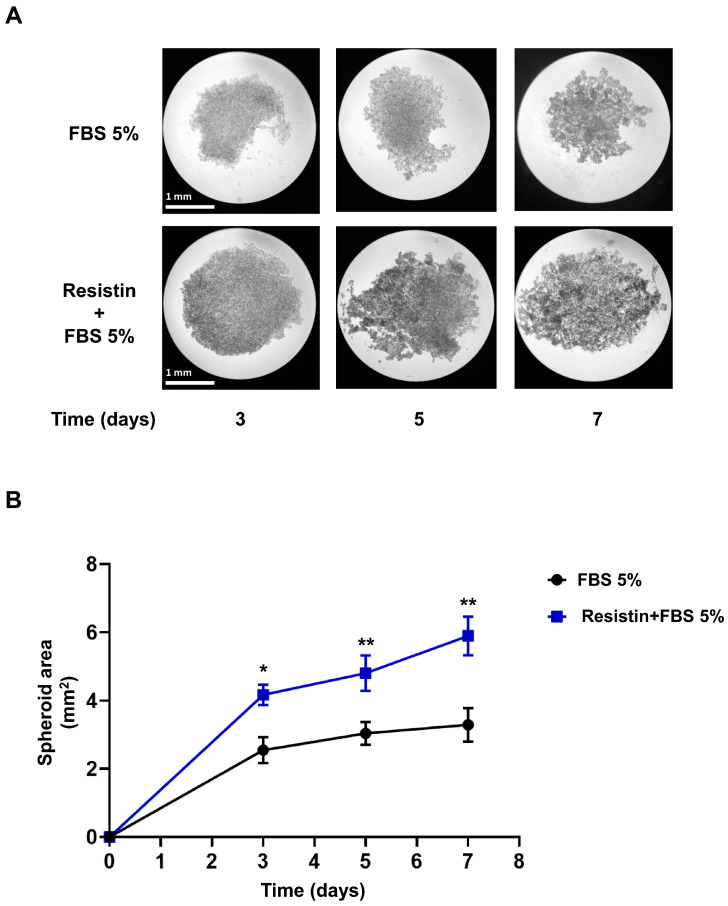
Resistin promotes an increase in the area of spheroids in PC3 cells. **(A)** 24-well plates were coated with 400 μL of 1.5% agarose. Subsequently, about 1 × 10^5^ PC3 cells were placed per well/condition, which were treated with 5% FBS alone or with 5% FBS + resistin (25 ng/mL) for seven days. During the assay, follow-up was maintained by taking photos of the spheroids on days 3, 5, and 7. Finally, the spheroidal area was analyzed with the ImageJ software. (B) The graph represents the mean ± SD of three independent experiments and is expressed as spheroid area (mm^2^). The scale bar is equal to 1 mm. * *p* < 0.05 and ** *p* < 0.01.

**Figure 4 life-13-02321-f004:**
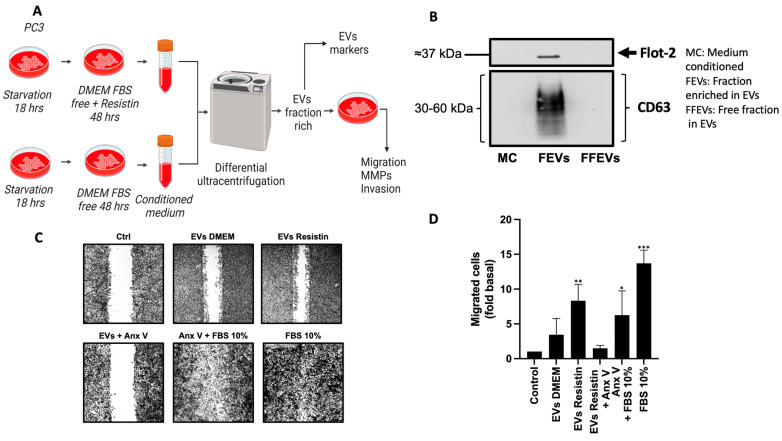
EVs derived from resistin-treated PC3 cells induce cell migration in PC3 cells. (**A**) Differential ultracentrifugation was used to isolate a fraction enriched with EVs. (**B**) EVs fraction was processed by Western blot assay using anti-flotillin-2 and anti-CD63 as EV markers; (**C**) cell migration was evaluated through scratch wound assay. Cells were treated with EVs derived from resistin-treated PC3 cells or without resistin treatment (30 µg/mL per condition). After 48 h, the migration of cells was analyzed. (**D**) Graphs are the mean ± SD of three independent experiments and are expressed as the fold value of migrated cells above the control value. * *p* < 0.05, ** *p* < 0.01, and *** *p* < 0.001.

**Figure 5 life-13-02321-f005:**
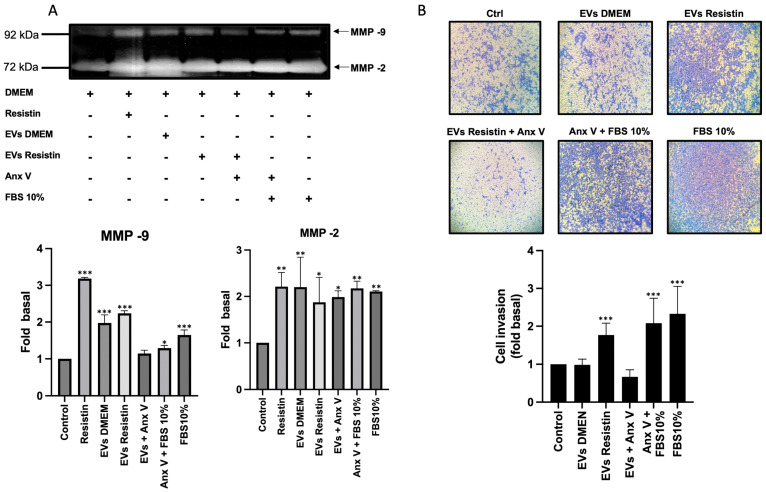
EVs derived from resistin-treated PC3 cells induce cell invasion in PC3 in an autocrine way. (**A**) PC3 cells were treated for 48 h with EVs derived from resistin-treated PC3 cells or without resistin treatment. Supernatants were collected and analyzed by zymography to evaluate MMP-2 and MMP-9 secretion; (**B**) Boyden chamber method was used to evaluate PC3 cell invasion when stimulated with EVs derived from resistin-treated PC3 cells or without resistin treatment. Cells were seeded in the upper chamber precoated with Matrigel, and treatment with EVs was added to the lower chamber. After 48 h, cell invasion was analyzed. The graph represents the mean ± SD of three independent experiments and is expressed as the fold of invasion above the control value. * *p* < 0.05, ** *p* < 0.01, and *** *p* < 0.001.

## Data Availability

The data obtained throughout the experiments can be provided by O.G.-H. upon reasonable request.

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
