# Peer review of "Resistin Induces Migration and Invasion in PC3 Prostate Cancer Cells: Role of Extracellular Vesicles"

_life, 2023, doi:10.3390/life13122321_

Round 1
Reviewer 1 Report
The manuscript entitled “Resistin Induces Migration and Invasion in PC3 Prostate Cancer Cells: Role of Extracellular Vesicles” by Oregel-Cortez et al., shows the Resistin-mediated promotion of migration and invasion over the prostate cancer cell line PC3. Moreover, the authors suggest that Resistin induces EVs release from the same cell line which could mediate the migration and invasion promotion.
The manuscript is well designed for the intended objectives, easy reading, though some issues should be addressed:
- The basal expression of Resistin in PC3 cells should be evaluated.
- The authors show that Resistin treatment induces migration and invasion. Then, they show similar effects for EVs from Resistin-treated PC3 cells. It would be good to show that there is no residual Resistin (from the treatment) in the last washing of EVs purification protocol.
- Manuscript should state if the used FBS is depleted of EVs.
- Figure 4A, should be better described in text. Not all the groups are explained and the observed results.
- Figure 4A, what is the proposed mechanism? I see an increase with Resistin, but I do not see a difference in EVs from Resistin or DMEM treated cells. Are the basal secreted EVs capable of promoting invasion?
- Line 357 (discussion section), change VEs by EVs.
- Please, add bibliographic reference to the EVs uptake inhibition by AnxV.
- In the discussion section, it would be nice to suggest a working hypothesis that contains all the presented data.
It is OK.
Reviewer 2 Report
The authors have drafted the manuscript called “Resistin Induces Migration and Invasion in PC3 Prostate Can-2 cer Cells: Role of Extracellular Vesicles.” They have identified that resistin promoted migration and invasion in PC3 cells in vitro. A couple of questions shall be addressed before further processing.
1. Vendor information shall be added for reagents and antibodies used.
2. Please specify the replicant numbers used for statistical analysis in the
manuscript. Is the control n =1? I cannot see the error bars for controls. Please correct me if I miss it.
3. Experimental details such as how much proteins were loaded for WB shall be included.
4. Adding quantitative and qualitative characterization of EVs can be helpful, such as morphology of EVs using electron microscopy, and NTA for the EVs count and size distribution.
Please check the grammar in this manuscript. For example, it shall be 8 x 106 rather than 8 x 106 for cell count. Line 127, the medium was centrifuged at 200 x g rather than to 200 x g. Inconsistencies among figure labels exist, please correct.
Reviewer 3 Report
In their study, Israel et al. demonstrated that resistin induces migration and invasion in PC3 Prostate Cancer Cells via extracellular vesicles. My overall interpretation is that biological mechanism info is missing. Adding one or two assays such as knockdown effect of resistin on various biomarkers to reveal the biological mechanism will add value to the paper. However, my comments are below:
Major Issues:
1. Western Blot in Figure 1B, is p-FAK normalized against Actin or FAK? Needs to show densitometry against FAK.
Raw images of western blots for FAK and p-FAK looks same, just a different exposure?
2. Figure 2A, please show raw images with different color scheme, unable to read labels on blue images.
3. In Figure 3, how 30ug/ml EVs concentration was quantified?
It would be interesting to see resistin presence in EVs. It could be possible that resistin in the EVs is activating signaling cascade in neighboring cells. Please add western blot for resistin in 3B
4. In Figure 4, MMP-9 is more in EVs resistin than EVs DMEM, which aligns with Figure3, however, MMP-2 is less in EVs resistin which should not be the case and so is Anx V+ FBS 10%. Any justification?
Minor issues:
1. Show molecular weights for biomarkers used in all the western blots
2. At multiple places 8 x 106 is written as 8 x 106, For e.g., Line 119,125
3. Line 139, 0.5ml of cold iced RIPA, if should be corrected
4. Figure 4B, top panel , image 5 should be labeled as Anx V+ FBS10%
Reviewer 4 Report
Oregel-Cortz et al present a paper on the role of the adipokine resistin in the migration and invasion of prostate cancer cell lines and the possible mediation of this effect by extracellular vesicles (EVs). There is some interesting data in the paper relating to MMPs but I think there are too many places where the story isn’t connected enough or detailed enough at this stage; but I think it could be.
Major
i) The whole study is done with a single cell line. I think this makes the conclusions rather narrow. There are many other good prostate cancer cell lines available and I think a comparison to LNCaP would be really interesting.
ii) EV isolation method refers to a previous paper, which refers to two other previous papers which refers to another set of previous papers. Given the importance of the EV work in this paper I think there needs to be a better description of how cells were cultured prior to EV isolation. Were they in serum free media? Or in EV-depleted serum containing media? Given the importance of the EV work in this paper I think EVs should also be confirmed by a nanosizer for particle size distributions? I think this is a significant omission and risky to just ask the reader to assume.
iii) Scratch-wound assay states that cell proliferation was inhibited by mitomycin C but I think the data showing no effect on proliferation should be in the supplementary. Presumably the authors confirmed there was no proliferation with a PrestoBlue or similar viability assay. Typically, scratch closure is quantified as a half-time for the wound closure. I am not sure why the authors go for a single time point but it is not typical for this type of assay.
iv) The quantification of the scratch closure is not described. Please do describe how the data in for example Figure 1A were quantified.
v) What is the reason for the inhibitory effect at higher resistin concentrations? Figure 1A shows quite clearly that 50 and 100 ng/ml were inhibitors of migration. What is the physiologically relevant concentration range of resistin?
vi) Invasion assays are done on a matrigel insert. The relevance of this composition to the ECM through which prostate cancer invade is questionable. How feasible are invasion assays with a more relevant ECM like mimic.
vii) The zymography data are missing recombinant MMP2 and MMP9 for validation. The supplied full size gels are not labelled in a way that lets any reader see which lane is which so this is very unhelpful and makes this data hard to judge.
viii) When EVs were added to cells (Figure 3) and an increase in the number of migrating cells was observed (3C). Was this dose dependent; i.e. the more EVs added gave a faster rate of closure?
ix) Assuming the EVs are the source of the MMPs that led to greater invasion could the authors have looked by western blotting for MMP2 and MMP9 in the EV fraction?
x) Figure 4 contains very contradictory data: the authors see in panel A no change in MMP2 or MMP9 levels from cells that received EVs from resistin-treated cells (4A) yet those same EVs appeared to enhance invasion in Figure 4B. How do the authors square this circle? Have the authors looked at levels of the MMP inducer EMMPRIN. There are recent reports that levels of EMMPRIN in EVs may be relevant in activating MMP2 rather than upregulating MMP2 expression.
Minor:
i) in many places in the methods there isn’t enough attention to detail in phrasing. So we see “8 x 106” cells in many places rather than the 6 being superscripted. We see “triton” instead of Triton, “EGT4” rather than EGTA, “SFB” instead of FBS and microMolar (symbols) to describe Boyden chamber pore size. I would recommend the authors check the methods most carefully for other small errors.
ii) Some other phrasing errors in the abstract where the very first sentence does not make sense and line 70 on page 2 also doesn’t read correctly. Line 203 contains “ok” rather than “of”.
iii) In several places EVs are called VEs
Round 2
Reviewer 1 Report
The authors have addressed all my concerns.
Author Response
The authors have addressed all my concerns.
Thank you for your invaluable suggestions.
Reviewer 2 Report
Manuscript is acceptable
Manuscript is acceptable
Author Response
Manuscript is acceptable
Thank you for your invaluable suggestions.
Reviewer 3 Report
Please screen manuscript one more time for minor mistakes , still at many places
>8 x 10^6 is written as 8 x 106.
>EVs is written as VEs.
Author Response
Please screen manuscript one more time for minor mistakes, still at many places
>8 x 10^6 is written as 8 x 106.
>EVs is written as VEs.
Thank you for your comment. The manuscript was reviewed and modified according to your suggestion.
Reviewer 4 Report
the authors have presented a detailed point-by-point commentary and improved the manuscript with this 2nd version. the new supplementary figures improve the manuscript considerably, particularly the dose-dependent effect seen in figure s2. I am still concerned that EVs are used without any characterisation/confirmation that they are EVs. Some level of basic characterisation is recommended by ISEV in their statement papers and I think without this the conclusion that EVs are mediating the effects remains unproven. I appreciate that technical constraints mean that they cannot validate their EVs but I think that somewhere in the paper the authors must acknowledge this limitation.
Author Response
The authors have presented a detailed point-by-point commentary and improved the manuscript with this 2nd version. The new supplementary figures improve the manuscript considerably, particularly the dose-dependent effect seen in figure s2. I am still concerned that EVs are used without any characterisation/confirmation that they are EVs. Some level of basic characterisation is recommended by ISEV in their statement papers and I think without this the conclusion that EVs are mediating the effects remains unproven. I appreciate that technical constraints mean that they cannot validate their EVs but I think that somewhere in the paper the authors must acknowledge this limitation.
Thank you for your comment. According to your suggestion, in the discussion section we have added the following information (page 11, line 406):
“Our data indicate that the EVs-enriched fraction presents Flot-2 and CD63, however, it is necessary to characterize the EVs populations by electron microscopy and/or nanoparticle tracking analysis”